# Catalysts of PtSn/C Modified with Ru and Ta for Electrooxidation of Ethanol

**Maria Aparecida Ribeiro Queiroz and Josimar Ribeiro \*** 

Departamento de Química, Universidade Federal do Espírito Santo, Av. Fernando Ferrari, 29075-910 Vitória, Brasil; mariaribeiro11@gmail.com
**\*** Correspondence: josimar.ribieor@ufes.br;Tel.: +55-27-4009-7948

**Abstract:** PtSn/C-type catalysts modified with Ta and Ru were prepared by the thermal decomposition of polymeric precursors with the following nominal compositions: $Pt_{70}Sn_{10}Ta_{20}/C$, $Pt_{70}Sn_{10}Ta_{15}Ru_5/C$, $Pt_{70}Sn_{10}Ta_{10}Ru_{10}/C$ and $Pt_{70}Sn_{10}Ta_5Ru_{15}/C$. The physicochemical characterization was performed by X-ray diffraction (XRD) and energy dispersive X-ray (EDX). The electrochemical characterization was performed using cyclic voltammetry, chronoamperometry and fuel cell testing. PtSnTaRu/C catalysts were characterized in the absence and presence of ethanol in an acidic medium ($H_2SO_4$ 0.5 mol $L^{-1}$). All the catalysts showed activity for the oxidation of ethanol. The results indicated that the addition of Ta increased the stability and performance of the catalysts, as the $Pt_{70}Sn_{10}Ta_{20}/C$ catalyst had the maximum power density of 27.3 mW $cm^{-2}$ in an acidic medium. The results showed that the PtSn/C-type catalysts modified with Ta and Ru showed good performance against alcohol oxidation, representingan alternative to the use of direct ethanol fuel cells.

**Keywords:** quaternary catalysts; tantalum; fuel cell

---

## 1. Introduction

Growing concern about environmental quality has led industries to seek cleaner technological alternatives and less toxic raw materials to reduce environmental impact and degradation. Together with the fear of the end of fossil fuels, the basis of the energy source and its price and supply crises, accompanied by policies of using clean and renewable energies, bring a context in which other means of energy production are used [1]. In this scenario, the principle of operation of fuel cells stands out because it presents a form of efficient generation of electric energy with little environmental impact [2–4].

In a fuel cell, the direct transformation of chemical energy into electric energy is achieved, where hydrogen gas fuel is the most prominent fuel used in these applications. However, the use of hydrogen as fuel presents some problems, such as lack of infrastructure for production, distribution and storage, as well as a high risk of flammability [2–4].

Due to the difficulties encountered in the use of hydrogen gas in fuel cells, the use of alcohols represents a good alternative for the production of clean energy [4–6]. Among the alcohols that can be used as fuels, ethanol stands out. It stands out as a renewable source and that can be produced on a large scale; furthermore, it has the advantages of low toxicity and easy storage [3,4,7]. However, ethanol, as well as other alcohols of two or more carbons [8–11], have disadvantages regarding their incomplete oxidation due to the difficulty in breaking the C–C bond, necessitating the use of platinum-based catalysts [3,8,9].

However, for oxidation reactions of carbonic molecules, due to the adsorption to the platinum active sites by intermediates formed during the reaction, the electrochemical activity of the platinum is reduced by poisoning during the oxidation process [10–12]. In order to improve the electrocatalytic

activity of platinum for the oxidation of organic molecules and to decrease the poisoning, other metals are added to platinum-based catalysts. For example, we found studies investigating the insertion of different metals such as Sn, Ru, Ni, Mo, Rh, Ga, Ti, Bi, Ir and W, among others, researched as co-catalysts to improve the properties of Pt-based catalysts [13–15].

Studies have shown that the PtSn/C catalyst presents excellent electrocatalytic activity in relation to ethanol oxidation [16–18]. The addition of Ru into PtSn/C catalysts has been also studied for alcohol oxidation [19–22]. In all these investigations the presence of Ru in the catalyst compositions increases the electrocatalytic activity toward alcohol oxidation. That increase in electrocatalytic activity when Ru is incorporated into the Pt-based catalysts is due to the bi-functional mechanism, where the contaminant (CO) preferentially adsorbs to the active sites when Pt is oxidized. When there is another metal, in this case ruthenium and tin, which are more oxidizable than platinum, these metals produce oxygenated species or hydrated oxide which act directly on the oxidation of the contaminant. In the specific case of Ru, as it undergoes oxidation at a potential 0.2 V, the bi-functional effect occurs because platinum breaks the O–H, C–H and C–C bonds [22,23].

Another metal that can act as a co-catalyst in platinum-based catalysts is tantalum. Studies have shown that binary catalysts of Pt and Ta improve the tolerance of CO [24]. Anwar et al. [25] used Ta and Ti oxide as supports for Pt catalysts which increased the durability compared to pure platinum and maintained their electrocatalytic activity. Masud et al. [26] obtained better results for the oxidation of methanol using platinum-based catalysts modified with Ta-oxide. The significant increase in methanol oxidation due to the addition of Ta as a co-catalyst together with Pt indicated a crucial role for $TaO_x$ in the process. The interaction between oxide and metal results in the electron donation of Pt to $TaO_x$, thus inducing a positive charge on the surface of Pt [26,27]. This induced positive charge weakens the bond strength of adsorbed CO at the sites of Pt, thereby increasing its oxidative removal ability and decreasing its surface coverage on the surface of Pt. Another mechanism through which Ta can improve the performance of Pt catalysts for the oxidation of ethanol is related to the high affinity of their oxides in relation to the OH species, providing species of oxygen at the surface of the catalyst with a lower potential compared to pure Pt catalysts. Thus, the presence of Ta perhaps favors the bi-functional mechanism and can accelerate the dissociation of $H_2O$ molecules to form $OH_{ads}$, facilitating the oxidation of adsorbed alcohols or poisons, and thus recovering the active sites of Pt so that the oxidation reaction continues [24,26].

Considering the improvements presented in platinum-based catalysts when adding Sn and Ru, we added Ta to PtSn/C- and PtSnRu/C-type catalysts, obtaining a ternary catalyst and three quaternary catalysts. The introduction of Ta in the catalysts was madeto evaluate whether the addition of this metal improves the electrocatalytic properties of the catalysts for the purpose of ethanol oxidation. The PtSnTaRu/C catalysts were obtained by the thermal decomposition of polymer precursors and their electrocatalytic activities were evaluated by cyclic voltammetry, chronoamperometry and fuel cell tests.

## 2. Results and Discussion

### 2.1. Physicochemical Characterizations

Figure 1 shows the X-ray diffraction (XRD) pattern of Ta- and Ru-modified PtSn/C catalysts supported on Vulcan XC72 carbon. The catalysts present diffraction peaks at 2θ = 39.8°, 46.2°, 67.5°, 81.4° and 85.7° relative to the diffraction planes (111), (200), (220), (311) and (222) of the platinum pattern [28] with face-centered cubic structure (fcc) [29]. The diffraction peaks relative to Sn, Ta and Ru in their isolated metallic form, or even their oxides, were not observed. However, it is not possible to state that they are not present, since these metals or their oxides may exist in very small amounts or an amorphous phase.

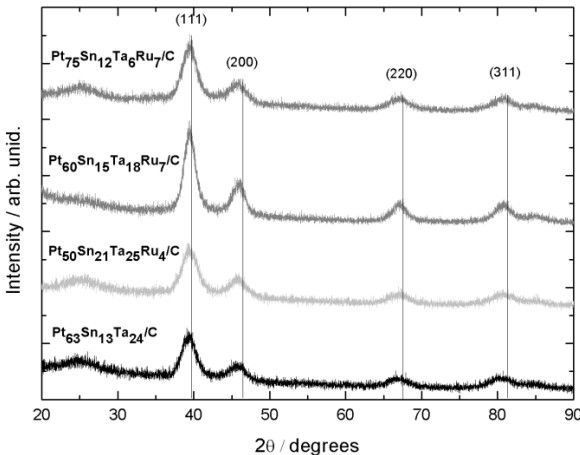

**Figure 1.** Of Ta- and Ru-modified PtSn/C catalysts supported on Vulcan XC72 carbon. Cu Kα ($\lambda$ = 1.5406 Å) at room temperature, 2θ = 10° to 90° with a step of 0.01° s$^{-1}$.

The catalysts exhibited a shift of diffraction peaks to lower values than the 2θ angles of the fcc Pt, indicating an increase in the lattice Pt parameter compared to the pure Pt (a = 3.920 nm), Table 1, probably due to the incorporation of the metals (Sn and/or Ta) in its structure [30]. This expansion of the lattice parameter due to the incorporation of Sn is already well known in the literature [31–34] since its atomic radius (Sn = 151 pm) is greater than that of platinum (Pt = 139 pm). Considering that the atomic radius of Tantalum (Ta = 200 pm) is also greater than that of Pt, this metal can also contribute to the increase of the lattice parameter, leading to the expansion of the crystalline lattice. From the XRD data, it was also possible to obtain the volume and average crystallite sizes (D) for each diffraction plane. For this, the equation of Scherrer [29] was used.

**Table 1.** Results of energy dispersive X-ray (EDX), X-ray diffraction (XRD) and electrochemical surface area (ECSA) obtained for PtSn/C catalysts modified with Ta and Ru.

| Nominal Composition (%mol) | Experimental Composition (%mol) [1] | a/Å | V/Å [3] | D [2]/nm | | | | Average Particle Size [3]/nm | ECSA-H$_2$ [4]/m$^2$ g$_{Pt}$$^{-1}$ |
|---|---|---|---|---|---|---|---|---|---|
| | | | | (111) | (200) | (220) | (311) | | |
| Pt$_{70}$Sn$_{10}$Ta$_{20}$/C | Pt$_{63}$Sn$_{13}$Ta$_{24}$/C | 3.941 ± 0.002 | 61.20 ± 0.08 | 4.3 | 4.6 | 3.4 | 2.9 | 5.5 | 14.6 |
| Pt$_{70}$Sn$_{10}$Ta$_{15}$Ru$_5$/C | Pt$_{50}$Sn$_{21}$Ta$_{25}$Ru$_4$/C | 3.944 ± 0.003 | 61.35 ± 0.15 | 3.9 | 3.7 | 2.9 | 3.6 | 3.8 | 8.7 |
| Pt$_{70}$Sn$_{10}$Ta$_{10}$Ru$_{10}$/C | Pt$_{60}$Sn$_{15}$Ta$_{18}$Ru$_7$/C | 3.942 ± 0.002 | 61.24 ± 0.11 | 5.3 | 5.2 | 4.8 | 4.7 | 4.8 | 10.7 |
| Pt$_{70}$Sn$_{10}$Ta$_5$Ru$_{15}$/C | Pt$_{75}$Sn$_{12}$Ta$_6$Ru$_7$/C | 3.944 ± 0.005 | 61.36 ± 0.28 | 4.2 | 3.7 | 3.3 | 3.4 | 5.0 | 7.5 |

[1] Results obtained by EDX; [2] Crystallite size calculated for each diffraction plane obtained by XRD; [3] Average particle sizes obtained by TEM; [4] ECSA calculated from the area of hydrogen obtained by cyclic voltammetry in an acidic medium.

Another parameter that allows the analysis of the formation of alloys is the volume of the unit cell of the Pt, with 60.380 Å$^3$ being the theoretical volume value [28]. The values shown in Table 1 indicate a probable formation of alloys due to the incorporation of metals (Sn, Ta and Ru) into the Pt structure crystalline.

The crystallite size values for the catalysts are consistent with the values already reported in the literature for catalysts based on Pt (2–7 nm) [21,35,36]. Because they are synthesized materials with the purpose of catalyzing the electrooxidation of alcohols, and since catalysis occurs on the surface of metals, it is possible to state that the smaller and more homogeneous the distribution of the particles in the material, the greater its surface area and the better its electrocatalytic activity [9,37].

Transmission electron microscopy (TEM) measurements were also performed to evaluate particle sizes and their frequencies, the results of which are shown in Figure 2 and Table 1. Analyzing the histograms of Figure 2, it is observed that, in general, all catalysts have more than 50% of particles ranging

in size from 3.5 to 5 nm, which can be confirmed by the average particle sizes for each catalyst shown in Table 1. In addition, Figure 2 shows that the particles of the quaternary catalysts (Figure 2B–D) are more homogeneously dispersed on the Vulcan XC72 carbon surface. However, from observations of the ternary catalyst (Figure 2A–C), it is possible to verify the formation of agglomerates of the nanoparticles, indicating a more heterogeneous material which may decrease the electrocatalytic activity of the material.

The energy dispersive X-ray measurements (EDX) data, Table 1, indicate that the experimental composition of the Ru and Ta modified PtSn/C catalysts were different from the nominal compositions, where the $Pt_{50}Sn_{21}Ta_{25}Ru_4$/C catalyst presented lower platinum loading and the $Pt_{75}Sn_{12}Ta_6Ru_7$/C catalyst had the highest load of platinum. The difference in the nominal and experimental compositions could be related to the preparation method in the synthesis of the catalysts, the thermal decomposition of polymeric precursors, also known as the Pechini method. This technique presents some deviation in the experimental proportions of the different metals when three or more components are inserted in the catalyst, so fluctuations in relation to the theoretical values are common [33,38]. The EDX data showed the presence of oxygen in all compositions: 6.4 at% for $Pt_{63}Sn_{12}Ta_{25}$/C, 8.5 at% for $Pt_{50}Sn_{21}Ta_{25}Ru_4$/C, 7.5 at% for $Pt_{60}Sn_{16}Ta_{18}Ru_7$/C and 5.8 at% for $Pt_{75}Sn_{12}Ta_6Ru_6$/C. The presence of oxygen in the catalysts suggests the formation of metallic oxide (SnOx, RuOx and TaOx).

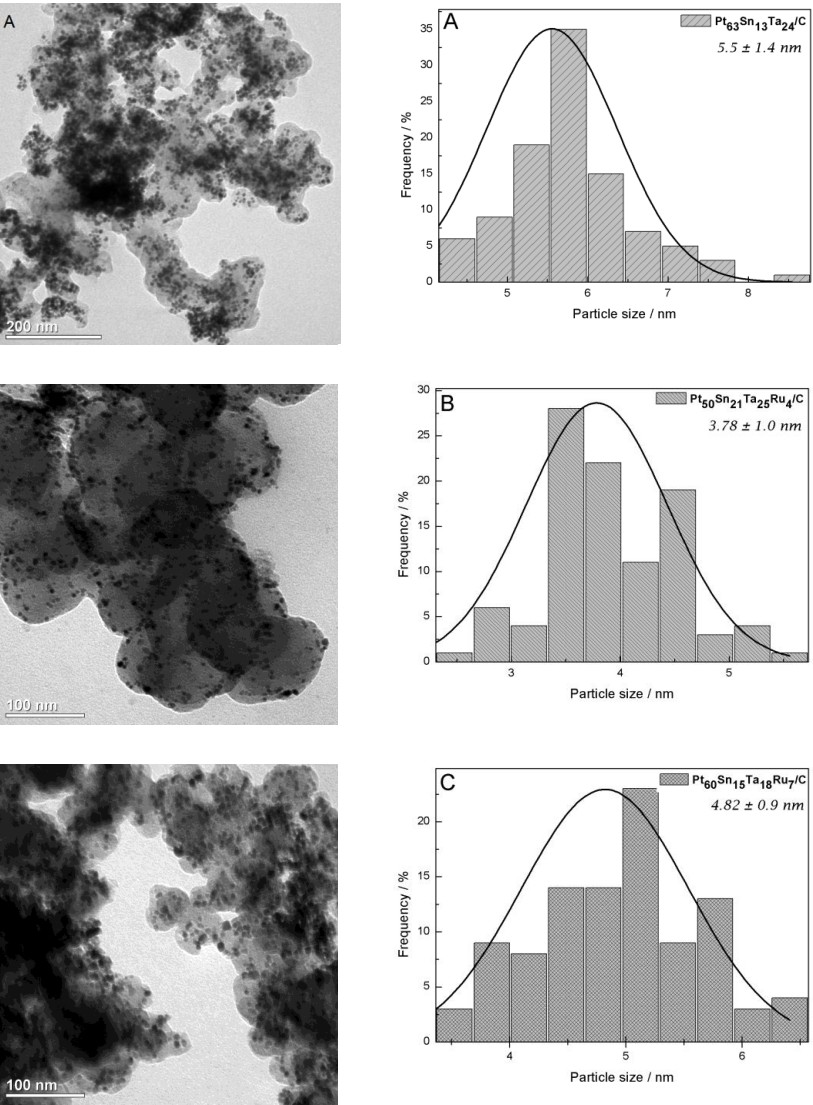

**Figure 2.** *Cont.*

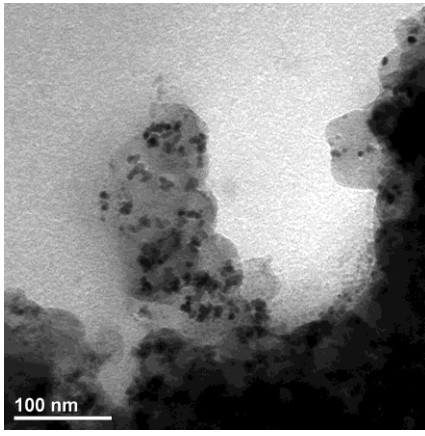 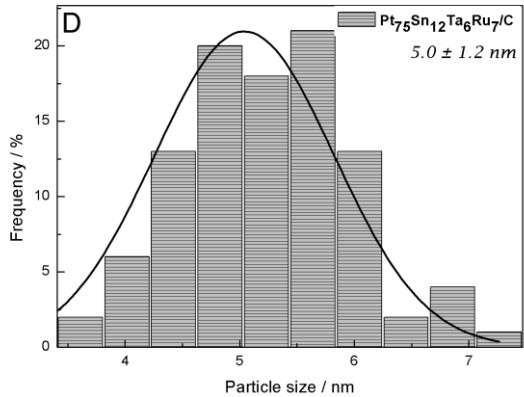

**Figure 2.** TEM images and histograms of particle size distributions of PtSn/C catalysts modified with Ta and Ru. (**A**) $Pt_{63}Sn_{13}Ta_{24}$, (**B**) $Pt_{50}Sn_{21}Ta_{24}Ru_4$, (**C**) $Pt_{60}Sn_{15}Ta_{18}Ru_7$ and (**D**) $Pt_{75}Sn_{12}Ta_6Ru_7$.

*2.2. Electrochemical Characterization*

The cyclic voltammetry (CV) of the PtSn/C catalysts modified with Ta and Ru (Figure 3) show similar behavior to that already presented by Pt-based catalysts on carbon [13,19,33,39]. The hydrogen adsorption/desorption region (−0.1 to 0.2 V vs. Ag/AgCl) is not well defined in relation to the pure Pt CV profile due to the possible formation of Sn oxides/hydroxides, Ru and Ta, which can block the Pt active sites [19,24,40]. This hypothesis is in agreement with the EDX data, which show that for all the catalysts the presence of oxygen leads to the formation of metallic oxide.

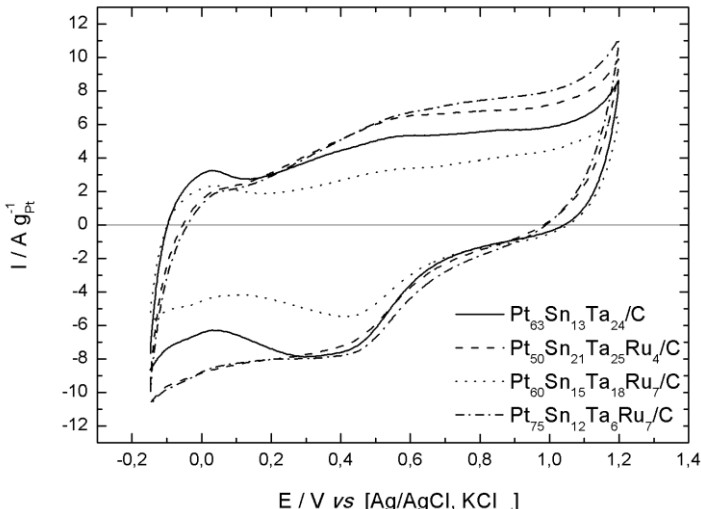

**Figure 3.** Cyclic voltammograms from PtSn/C catalysts modified with Ta and Ru, supported on Vulcan XC72 carbon, in supporting electrolyte ($H_2SO_4$ 0.5 mol $L^{-1}$) with a scan rate of 20 mV $s^{-1}$ (normalized current by the mass of Pt in grams).

All catalysts showed an increase in current density from 0.15 V vs. Ag/AgCl where the charge of the double layer became more evident. This behavior has also been observed in other Pt-based catalysts supported on carbon modified with transition metals [12,18,26,38,41], and the presence of Sn and Ru metals have already been related to the increase of the double layer charge [19,42,43].

The ternary catalyst ($Pt_{63}Sn_{13}Ta_{24}/C$) presented the highest electrochemical surface area (ECSA) value (Table 1). This result indicates that the addition of Ta in PtSnRu/C catalysts does not promote the increase of ECSA, although Ru and Ta could contribute to the increase in the electrocatalytic activity of Pt through the bi-functional mechanism [22–26].

Comparing the ECSA values obtained with the experimental composition of quaternary catalysts, a relation between the ratio of Sn and Ta could be observed. For the lower ratio of $Sn_{15}/Ta_{18} = 0.83$, a higher ECSA value was observed (10.7 m$^2$ g$_{Pt}$$^{-1}$) and for the higher ratio of $Sn_{12}/Ta_6 = 2.0$, a lower ECSA value was obtained (7.5 m$^2$ g$_{Pt}$$^{-1}$). The ECSAvalues obtained in this paper were higher than those of other binary and ternary catalysts with high Pt content (70 to 100 at%) [38,44].

The cyclic voltammograms in ethanol solution 1.0 mol L$^{-1}$ in supporting electrolyte are shown in Figure 4.

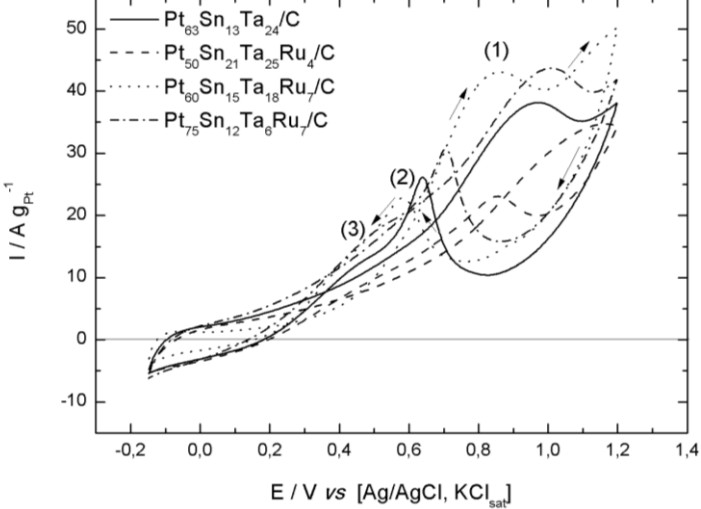

**Figure 4.** Cyclic voltammograms from PtSn/C catalysts modified with Ta and Ru in the presence of ethanol 1.0 mol L$^{-1}$ in H$_2$SO$_4$ 0.5 mol L$^{-1}$ with a scan rate of 20 mV s$^{-1}$.

The absence of desorption/adsorption hydrogen peaks indicates that the adsorption of ethanol molecules on Pt active sites is stronger than that of hydrogen molecules [30,32]. As the potential increases, there is an increase in current associated with the oxidation of ethanol, leading to the formation of different intermediates [1,45].

The peak current ($I_p$) is an important parameter to evaluate the catalysts for ethanol oxidation. The values of $I_p$ are related to kinetic factors of the electrocatalytic reactions and electronic transfer rate, where a higher $I_p$ indicates a higher ethanol oxidation rate [3,32,46]. The onset ethanol oxidation potential ($E_{onset-Ethanol}$) is another important parameter to be evaluated. $E_{onset-Ethanol}$ is related to the bi-functional mechanism as well as the thermodynamics of the reactions [26,33,47]. The insertion of other metals and their oxides in Pt-based catalysts could supply oxygen species at low potentials that can help in ethanol oxidation [26,46,48]. In this way, the decrease of the $E_{onset-Ethanol}$ can be indicative of a good catalyst. Table 2 shows the values of $I_p$ and the potentials obtained for the catalysts.

**Table 2.** Currents and potentials of the oxidation peaks for the PtSn/C-type catalysts modified with Ta and Ru.

| Catalysts | $E_{onset-Ethanol}$ | (1) | | (2) | | (3) | |
|---|---|---|---|---|---|---|---|
| | E/V | E/V | $I_p/Ag_{Pt}^{-1}$ | E/V | $I_p/Ag_{Pt}^{-1}$ | E/V | $I_p/Ag_{Pt}^{-1}$ |
| **Pt$_{63}$Sn$_{13}$Ta$_{24}$/C** | 0.26 | 0.97 | 38.11 | 0.63 | 25.80 | 0.45 | 12.95 |
| **Pt$_{50}$Sn$_{21}$Ta$_{25}$Ru$_4$/C** | 0.33 | 1.16 | 34.64 | 0.86 | 23.08 | 0.47 | 9.23 |
| **Pt$_{60}$Sn$_{15}$Ta$_{18}$Ru$_7$/C** | 0.36 | 0.86 | 42.80 | 0.58 | 22.83 | 0.46 | 15.91 |
| **Pt$_{75}$Sn$_{12}$Ta$_6$Ru$_7$/C** | 0.20 | 1.00 | 43.55 | 0.70 | 30.75 | 0.46 | 15.91 |

The cyclic voltammograms of Figure 3 and the data shown in Table 2 indicate that Pt$_{75}$Sn$_{12}$Ta$_6$Ru$_7$/C had the highest $I_p$ value. When comparing the $E_{onset-Ethanol}$ values, Pt$_{63}$Sn$_{13}$Ta$_{24}$/C

and $Pt_{75}Sn_{12}Ta_6Ru_7/C$ were found to show the lowest values. The reduction of $E_{onset-Ethanol}$ can be explained by the bi-functional mechanism [49,50] which may be due to the addition of more easily oxidizable metals in the platinum structure [22–26]. Oxides of Ta or Ru can aid in the oxidation of ethanol due to the supply of oxygenated species that increases the affinity of the catalyst and consequently the oxidation of the ethanol due to the oxidation of adsorbed species on the active sites of Pt, thus renewing the active sites for new ethanol molecules to be oxidized [22,26].

After the measurements of CV in the presence of ethanol 1.0 mol $L^{-1}$ in $H_2SO_4$ 0.5 mol $L^{-1}$, the catalysts were subjected to experiments at a fixed potential, where 0.4 V vs. Ag/AgCl was applied for 2 h.

Electrolysis was used to evaluate the catalytic performance of the catalysts and their behavior in ethanol by applying a constant potential of 0.4 V. As the ethanol oxidation reaction advances, intermediates accumulate on the catalyst surface, thus preventing new ethanol molecules from being oxidized and reducing the current over time [33]. For all the catalysts, the current density decreased abruptly in the first 5 min in which the reaction took place, as shown in Figure 5. At 5 min, the $Pt_{63}Sn_{13}Ta_{24}/C$ catalyst showed the lowest drop and the highest current density value, around 8.3 A $g^{-1}_{Pt}$, while the others had values between 6 and 7 A $g^{-1}_{Pt}$. Between 10 and 50 min, a small reduction was observed in the current density and after 50 min it decreased even further. This decrease in the current densities can occur due to the instability of the metal surface caused by phenomena such as the segregation, crystallization and agglomeration of particles, leading to the degradation of the catalytic material [34,38]. However, one can infer that from the 10 min timepoint, the current density is almost stable. This stabilization in the current density values may indicate an improvement in the catalytic activity of the material through the bi-functional mechanism or the electronic effect. As the added metals (Ru and Ta) modify the electronic structure of Pt [14,26] and promote interfacial water activation by enriching the surface of the material with $OH_{ads}$, this weakening the connection between the intermediates of the reaction of ethanol oxidation with the active Pt sites. Thus, the stabilization of the current observed here may be an indication that the catalysts were able to renew the active Pt sites, indicating improved catalytic activity and therefore showing high values of current density over time.

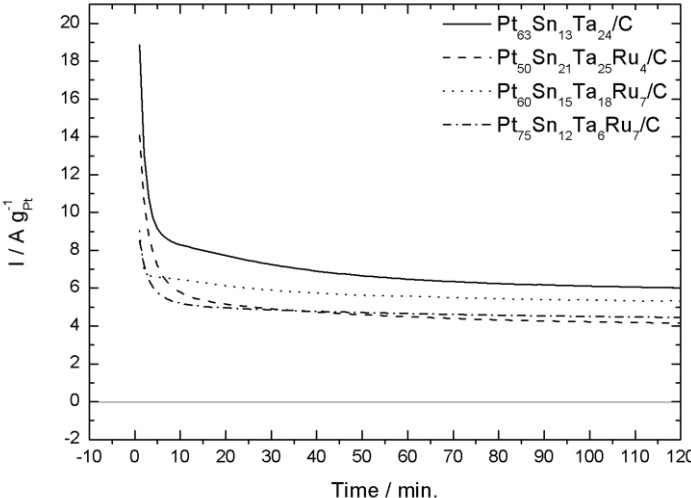

**Figure 5.** Chronoamperometry in ethanol 1.0 mol $L^{-1}$ in $H_2SO_4$ 0.5 mol $L^{-1}$ with a potential of 0.4 V vs. Ag/AgCl for 2 h.

In this sense, it is interesting to evaluate the current densities of the material and consequently the chronoamperogram over time in the presence of ethanol. It is thus possible to evaluate the stability of the material. At the end of 2 h, $Pt_{63}Sn_{13}Ta_{24}/C$ presented the highest value of current density, around 6.1 A $g^{-1}_{Pt}$. On the other hand, the quaternary catalysts showed similar values of current density and curve behavior, with values close to each other—in the range of 4.2 to 5.3 A $g_{Pt}^{-1}$ at the end of 2 h.

Another fact to be observed stems from the small variation in the current throughout the experiment, indicating that the quaternary compositions presented better stability than the ternary catalysts. For example, the composition $Pt_{75}Sn_{12}Ta_6Ru_7/C$ showed the lowest variation in current density: 6.48 A $g_{Pt}^{-1}$ in the first 10 min and 5.4 A $g_{Pt}^{-1}$ at the end of the experiment. These values are higher than those of PtSn/C [38] binary catalysts with the same Pt content, 70% mol and PtRu/C [19] with 45% metal. Although again the ternary catalyst, $Pt_{63}Sn_{13}Ta_{24}/C$, presented the best results among the compositions studied in this paper, when comparing the results of ternary catalyst PtSnRu/C [20,21], the currents obtained by the quaternary catalysts herein were superior. This indicates that the addition of Ta to the base composition PtSnRu/C increased the electrocatalytic activity.

The catalysts were tested in an ethanol fuel cell using in the first test only ethanol 2.0 mol $L^{-1}$ and in the second test ethanol 2.0 mol $L^{-1}$ in $H_2SO_4$ 0.5 mol $L^{-1}$ to evaluate the influence of the supporting electrolyte on the electrocatalytic activity of the materials. Figure 6 shows the electrical performance of a 5 cm$^2$ direct ethanol fuel cell (DEFC) at 90 °C using the Pt-based catalysts.

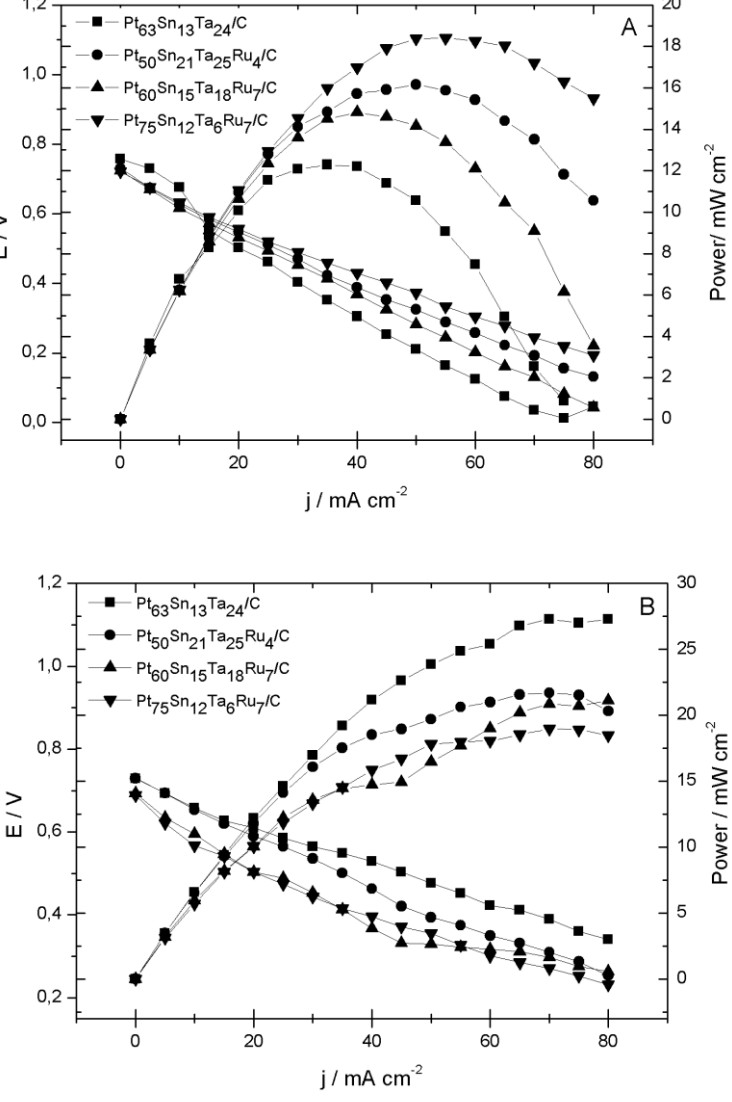

**Figure 6.** Ethanol fuel cell tests, operating at 90 °C with a fuel flow of 2.0 mL min$^{-1}$, $O_2$ pressure of 3 bar and pressure of 1 bar of ethanol with a metal loading of 2.0 mg cm$^{-2}$ in the anode and cathode of Pt, 2.0 mg$_{Pt}$ cm$^{-2}$, for the different catalysts. (**A**) Ethanol 2.0 mol $L^{-1}$, (**B**) ethanol 2.0 mol $L^{-1}$ in $H_2SO_4$ 0.5 mol $L^{-1}$.

The open circuit potential is an important measure because it can give an indication of the performance of the material when used in fuel cells. Since the standard potential for the electrooxidation of ethanol in fuel cells is 1.145 V, the closer the potential is to this value, the better the performance of the catalyst as an anode in an ethanol cell [51]. As seen in Figure 6A, 0.76 V was the highest open circuit potential recorded for the $Pt_{63}Sn_{13}Ta_{24}/C$ catalyst; the lowest value was 0.72 V for $Pt_{50}Sn_{21}Ta_{25}Ru_4/C$ and $Pt_{60}Sn_{15}Ta_{18}Ru_7/C$. When in an acidic medium(Figure 6B), potentials were also close: 0.73 V for $Pt_{63}Sn_{13}Ta_{24}/C$ and 0.69 V for $Pt_{50}Sn_{21}Ta_{25}Ru_4/C$ and $Pt_{60}Sn_{15}Ta_{18}Ru_7/C$. Comparing the fuel cell test results obtained with ethanol (Figure 6A), one can observe that the $Pt_{75}Sn_{12}Ta_6Ru_7/C$ catalyst presented the highest recorded power density value, in agreement with the results of ethanol voltammetry (Figure 4 and Table 2), where this composition presented the highest values of current density, lower values of $E_{onset-ethanol}$ and the lowest current variation over time in chronoamperometry (Figure 5).

Figure 7 shows the normalized maximum power values obtained for the PtSnTaRu/C catalyst. It is possible to verify the difference in efficiency of each catalyst in the fuel cell tests in relation to the maximum power produced by mass platinum. In absence of sulfuric acid, the maximum power density values of the DEFC were normalized with respect to the Pt loading. It was found that the electrical performance of the PtSnTaRu/C catalysts increased with the presence of Ru in the catalyst composition. On the other hand, in presence of sulfuric acid, the catalysts showed the opposite tendency.

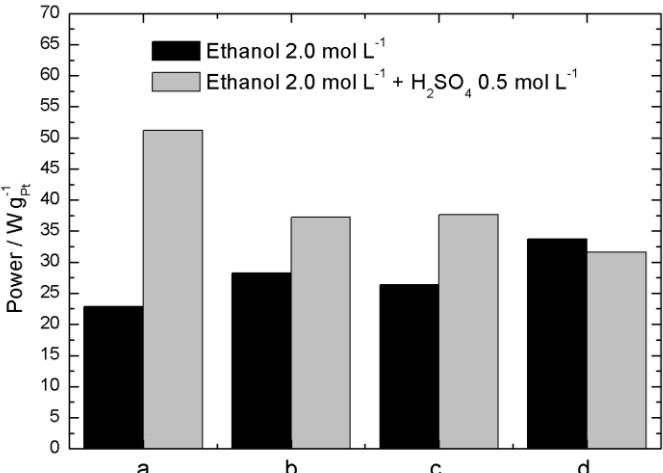

**Figure 7.** Normalized maximum power values obtained by mass of platinum in the ethanol fuel cell test for the catalysts under different conditions. (a) $Pt_{63}Sn_{13}Ta_{24}/C$; (b) $Pt_{50}Sn_{21}Ta_{25}Ru_4/C$; (c) $Pt_{60}Sn_{15}Ta_{18}Ru_7/C$; (d) $Pt_{75}Sn_{12}Ta_6Ru_7/C$.

It is noteworthy that the open circuit potential values obtained for all catalysts and in both conditions were higher than that presented by pure platinum (0.5 V) [16,52].Moreover, they were similar to those exhibited by binary PtRu/C catalysts (0.7 V) [19,53,54], but lower than those shown by PtSn/C binaries (0.8 V) [19,53]. The power densities were also higher compared to pure platinum (7–10 mW cm$^{-2}$) [16,53] catalysts, as well as PtRu/C (8–30 W g$^{-1}_{Pt}$) [19,52] and PtSnRu/C (17 W g$^{-1}_{Pt}$) [52] ternary catalysts. Table 3 shows the open circuit potential and power density for comparison purposes.

**Table 3.** Open circuit potential and power density values of different catalysts found in the literature.

| Catalyst | Loading Pt | Open Circuit Potential | Power Density | Ref. |
|---|---|---|---|---|
| **Pt/C** | 100% | 0.55 V | 10 mW cm$^{-2}$ | [16] |
|  |  | 0.48 V | 7 mW cm$^{-2}$ | [53] |
| **PtSn/C** | 80 wt% | 0.75 V | 15 W g$_{Pt}^{-1}$ | [52] |
|  | 75 wt% | 0.78 V | 27 mW cm$^{-2}$ | [42] |
|  | 70 wt% | 0.70 V | 25 mW cm$^{-2}$ | [53] |
|  | 45 wt% | 0.83 V | 32 W g$_{Pt}^{-1}$ | [19] |
| **PtRu/C** | 70 wt% | 0.60 V | 8 W g$_{Pt}^{-1}$ | [52] |
|  | 50 wt% | 0.70 V | 30 mW cm$^{-2}$ | [54] |
|  | 45 wt% | 0.75 V | 30 W g$_{Pt}^{-1}$ | [19] |
| **PtSnRu/C** | 30 wt% | 0.6 V | 17 W g$_{Pt}^{-1}$ | [52] |

## 3. Materials and Methods

### 3.1. Support Functionality

A chemical functionalization with $HNO_3$ was carried out in order to modify the surface of the support and increase the adsorption of the metals. For this, 2 g of carbon (Vulcan XC72, Carbot, Billerica, MA, USA) together with 50 mL of ultrapure water (SARTORIUS mini, Arium®, Göttingen, Germany) with a resistivity of 18.2 MΩ cm at 22 °C) was heated to boiling for 1 h and then filtered with the aid of a pump and washed with ultrapure water at room temperature. The filtered carbon was added to $HNO_3$ (Sigma-Aldrich, Cotia, SP, Brazil) 1.0 mol L$^{-1}$ solution and boiled for 30 min followed by filtration. The filtered carbon was oven-dried at 100 °C for 4 h and annealed in an oven (QUIMIS, Q318M, Diadema, SP, Brazil) at 400 °C for 1 h in atmospheric air.

### 3.2. Preparation of PtSnTa/C and PtSnTaRu/C Catalysts

The PtSn/C catalysts modified with Ta and Ru were synthesized by the method of thermal decomposition of polymeric precursors (DPP) [13,33,55,56].

The metal precursor resins were prepared separately by the mixture of citric acid (CA) and ethylene glycol (EG), following the molar ratio of 1:4 CA:16 EG. Initially 1.28 g of CA (Sigma-Aldrich, Cotia, SP, Brazil) and 10 mL of EG (Sigma-Aldrich, Cotia, SP, Brazil) were heated in a heating plate (DiagTech DT3110H, São Paulo, SP, Brazil) until complete dissolution of the CA, 60–65 °C. Without increasing the temperature, the solution of metal ions in isopropanol (NEON, Rio de Janeiro, RJ, Brazil) with a concentration of 0.05 mol L$^{-1}$ was added dropwise. After the complete addition of the metal ion solution, the system was heated under stirring to 80–95 °C for 1–2 h for the esterification and evaporation of excess isopropanol. After this process, metal resins were obtained with the following concentrations: Pt resin (1.51 × 10$^{-4}$ mol of Pt g$^{-1}$); Sn resin (7.34 × 10$^{-5}$ mol of Sn g$^{-1}$); Ta resin (1.23 × 10$^{-4}$ mol of Ta g$^{-1}$); Ru resin (1.57 × 10$^{-4}$ mol of Ru g$^{-1}$). The salts of the metals used in the preparation of the resins were: $H_2PtC_6$ (Sigma-Aldrich, Cotia, SP, Brazil); $SnCl_2$ (Sigma-Aldrich, Cotia, SP, Brazil); $C_{10}H_{25}O_5Ta$ (Sigma-Aldrich, Cotia, SP, Brazil); $RuCl_3 \cdot xH_2O$ (Sigma-Aldrich, Cotia, SP, Brazil).

One hundred milligrams of each catalyst were prepared, 60 mg (60 wt% carbon) of which corresponds to the functionalized carbon support and 40 mg (40 wt% metal) to the total metal amount. The required carbon and quantities of the resins were placed in a 5 mL glass vial along with 1.0 mL of ethanol (Sigma-Aldrich, Cotia, SP, Brazil) to facilitate the dispersion of materials. The resulting mixture was placed in an ultrasonic bath (UNIQUE USC-1400, Indaiatuba, SP, Brazil) for 30 min for homogenization followed by 24 h in an oven (QUIMIS Q317M-22, Diadema, SP, Brazil) at 60 °C for solvent evaporation and polymerization. The catalysts were obtained after calcination at 350 °C in an oven (QUIMIS Q318M, Diadema, SP, Brazil) for 30 min under ambient atmosphere. Four catalysts

having the following nominal compositions were synthesized: $Pt_{70}Sn_{10}Ta_{20}/C$, $Pt_{70}Sn_{10}Ta_{15}Ru_5/C$, $Pt_{70}Sn_{10}Ta_{10}Ru_{10}/C$ and $Pt_{70}Sn_{10}Ta_5Ru_{15}/C$.

### 3.3. Physicochemical Characterization

The XRD analysis was performed using a Bruker D8 diffractometer (Karlsruhe, Germany) operating with Cu $K\alpha$ radiation (1.5406 Å), with a $2\theta$ scan of 10° to 90° (0.01° min$^{-1}$). The crystallite size was estimated using the equation Scherrer [29] (1) for all the diffraction planes:

$$D = \frac{K\lambda}{\beta cos\theta_\beta} \tag{1}$$

where $D$ corresponds to the apparent size of the crystallite, $K$ to the form factor (0.9 for spherical crystallites), $\lambda$ to the wavelength of the radiation, $\beta$ to the diffraction full width at half-maximum intensity (FWHM) and $\theta_\beta$ to the angle at maximum intensity and the wavelength.

The lattice and volume parameters of the unit cell were determined using the least squares method via U-Fit computational software. The experimental values of $2\theta$ and the reflection planes (hkl) were used to obtain unit cell parameters.

The EDX measurements were performed in order to obtain the experimental metal compositions of each catalyst. Measurements were obtained using a Carl Zeiss scanning electron microscope (model EVO 01, Jena, Germany) coupled to an energy dispersive X-ray detector.

Transmission electron microscopy (TEM) measurements were obtained using a JEOL JEM-1400 (Peabody, MA, USA) Electronic Microscope, with a magnification of 800 k and 120 kV voltage.

### 3.4. Electrochemical Measurements

All solutions were prepared using ultrapure water (SARTORIUS mini, Arium®, Goettingen, Germany) with resistivity of 18.2 MΩ cm at 22 °C). For the electrochemical measurements, a 150 mL electrochemical cell was used with an Ag/AgCl reference electrode (ANALION, R682A, Ribeirão Petro, São Paulo, Brazil.) with saturated KCl solution, a counter electrode of carbon with an area of 3.15 cm$^{-2}$ and a working electrode (ANALION, K5705, Ribeirão Petro, São Paulo, Brazil) with an area of 0.066 cm$^{-2}$. A solution of each catalyst containing 1.00 mg of material was prepared with 5 μL Nafion® (Sigma-Aldrich, Cotia, SP, Brazil) and 95 μL ethanol (Sigma-Aldrich, Cotia, SP, Brazil). This solution was placed in an ultrasonic bath for 30 min. The final solution, 100.0 μL, was deposited on the pre-polished vitreous carbon working electrode with 0.3 μm alumina (SKILL-TEC, São Paulo, SP, Brazil) and dried; finally, the electrode was dried at room temperature for at least 30 min.

Cyclic voltammetry (CV) was used to activate the catalytic sites of the catalysts with 50 cycles and a scan rate of 50 mV s$^{-1}$ in the supporting electrolyte $H_2SO_4$ (Sigma-Aldrich, Cotia, SP, Brazil) 0.5 mol L$^{-1}$solution. This electrode conditioning procedure was performed for all electrochemical measurements. After electrochemical conditioning, two cycles were recorded with a scan rate of 20 mV s$^{-1}$in an acidic medium and in the presence of ethanol (Sigma-Aldrich, Cotia, SP, Brazil) 1.0 mol L$^{-1}$ in $H_2SO_4$ 0.5 mol L$^{-1}$from −0.15 to 1.2 V versus Ag/AgCl. Before each measurement the supporting electrolyte was purged with nitrogen gas for 15 min for the deoxygenation of the solution.

The ECSA [44,57,58] of each catalyst was obtained, assuming that the reference charge for the oxidation of a hydrogen monolayer on a platinum electrode corresponds to 2.1 C m$^{-2}$. For the calculation of ECSA, Equation (2) was employed:

$$ECSA\left(m^2 g_{Pt}^{-1}\right) = \frac{Q_H}{2.1 x m_{Pt}} \tag{2}$$

where $Q_H$ is the hydrogen desorption charge (Coulombs) and $m_{Pt}$ is the amount of platinum dispersed in the volume of electrocatalytic paint deposited on the surface of the working electrode [57,58]. The value of $Q_H$ can be obtained from of the area of the voltammetric profile in the an acidic medium

in the potential range between $-0.15$ V and 0.10 Vvs. Ag/AgCl (the scan rate used in the measurement was 20 mV s$^{-1}$). The chronoamperometry (CA) measurements were performed in the presence of ethanol 1.0 mol L$^{-1}$ in H$_2$SO$_4$ 0.5 mol L$^{-1}$vs. Ag/AgCl at fixed potential of 0.4 V vs. Ag/AgCl for 2 h. All measurements of cyclic voltammetry and chronoamperometry were performed using a Potentiostat/GalvanostatVersaSTAT 4, PAR-Ametek (Oak ridge, TN, USA) in ambient temperature between 22 and 24 °C.

For the fuel cell test, the membrane electrode assemblies (MEA) consisting of an anode/Nafion® membrane/cathode was mounted. The anode was prepared with a diffuser medium for the catalytic material consisting of 5.0 cm$^2$ carbon tissue. The loadings of the catalysts were deposited to obtain 2.0 mg cm$^{-2}$of metal. For this purpose, solutions of the catalysts where the loadings were equal to 10 mg of Pt were added and ethanol and Nafion® (Sigma-Aldrich, Cotia, SP, Brazil) were added in the following ratio: 1.0 mg of catalyst to 95 μL of ethanol and 5 μL of Nafion®. The solution was placed in an ultrasonic bath (UNIQUE USC-1400, Indaiatuba, SP, Brazil) for 30 min. Then the solution was deposited on carbon tissue (25.0 by 25.0 μL) until the entire tissue was covered. The anode was placed in an oven (Quimis Q317M-22, Diadema, SP, Brazil) at 60 °C for solvent evaporation for 30 min. The hydrated Nafion® membrane [59] was cut to yield 36.0 cm$^2$. For the cathode, a commercial tissue (FuelCellsEtc MLGDE, Providence Village, TX, USA) was used with a 5.0 cm$^2$ area composed of 2.0 mg cm$^{-2}$ of Pt nanoparticles. The set formed by the anode/membrane Nafion®/cathode was pressed at 130 °C and 1 ton for 3 min in a hydraulic press (CARVER 4389, Wabash, IN, USA). Then it was placed between the graphite plates of the fuel cell (ELECTROCELL FC25-1M, São Paulo, SP, Brazil) and the assembly was screwed together. The test was carried out with the help of the cell test station (ELECTROCELL model CDE 50A-2V, São Paulo, SP, Brazil) equipped with dynamic load. During each measurement the flow of the ethanol solution of 2.0 mL s$^{-1}$ was maintained at the anode by a peristaltic pump (MSTECNOPON LDP-101-3, Piracicaba, SP, Brazil) in the absence and presence of the supporting electrolyte, H$_2$SO$_4$ 0.5 mol L$^{-1}$. At the cathode, the flow of oxygen gas was hydrated and heated to 90 °C. Throughout the experiment the pressure was maintained at approximately 3.0 bar with an oxygen flow of 110 mL min$^{-1}$ and a cell temperature of 90 °C.

## 4. Conclusions

The results of the XRD analysis indicate that the incorporation of metals, specifically Sn, Ru and Ta atoms in the face-centered cubic Pt structure, lead to an increase in the lattice parameter relative to pure Pt. Furthermore, the crystallite sizes range from 2.9 to 5.3 nm and they are close to the value considered ideal for this type of material (2–7 nm). The TEM data of catalysts show good dispersion in the carbon support and the EDX indicate that there was oxygen present in all the catalysts investigated in this paper.

In the electrochemical measurements, the PtSnTaRu/C catalysts showed an increase in electrocatalytic activity regarding the ethanol oxidation. The quaternary Pt$_{75}$Sn$_{12}$Ta$_6$Ru$_7$/C catalyst showed better results in cyclic voltammetry in the presence of ethanol with the lowest E$_{onset-ethanol}$ value as well as the highest current density peak, indicating that the addition of Ta improved the electrocatalytic activity. Furthermore, the chrono-data of PtSnTaRu/C catalysts showed great catalytic surface renewal capacity because the current density was very stable throughoutalmost the entire experiment measurement time period.

In the fuel cell test, this same composition also presented the best performance compared to the other catalysts synthesized in the paper. When the cell test was conducted just with 2.0 mol L$^{-1}$ ethanol, it was verified that the presence of Ta in the catalysts could promote their performance. The results indicated that addition of Ta in PtSnRu/C catalysts increased the ethanol oxidation capacity of catalysts compared to pure Pt/C, PtRu/C and PtSnRu/C catalysts. On the other hand, when the fuel cell test was conducted using ethanol in an acidic medium (0.5 mol L$^{-1}$ H$_2$SO$_4$), the power densities were higher (18–27 mV cm$^{-2}$) than those found when the test was conducted only in ethanol

(12–19 mV cm$^{-2}$). In addition to the improved performance of the materials in terms of power density, it was observed that in an acidic medium the tendency for Ta content was reversed.

Based in all the results obtained in this investigation, one can infer that the presence of Ta and Ru improve the catalytic activity of PtSn/C-based catalysts.

**Author Contributions:** M.A.R.Q. and J.R. contributed equally.

**Funding:** This research was funded by Coordenação de Aperfeiçoamento de Pessoal de Nível Superior—Brasil (CAPES); Fundação de Amparo à Pesquisa e Inovação do Espírito Santo (FAPES); Conselho Nacional de DesenvolvimentoCientífico e Tecnológico (CNPq); Universidade Federal do Espírito Santo (UFES) and Vale S.A.

**Acknowledgments:** We gratefully acknowledge the funding of this study received from Coordenação de Aperfeiçoamento de Pessoal de Nível Superior—Brasil (CAPES); Fundação de Amparo à Pesquisa e Inovação do Espírito Santo (FAPES); Conselho Nacional de Desenvolvimento Científico e Tecnológico (CNPq); Universidade Federal do Espírito Santo (UFES) and Vale S.A. We are grateful to Marco C.C. Guimarães for the help with the TEM measurements.

**Conflicts of Interest:** The authors declare no conflict of interest.

## Nomenclature

| | |
|---|---|
| DAFC | Direct alcohol fuel cell |
| XRD | X-ray diffraction |
| D | Apparent size of the crystallite |
| K | Shape factor |
| λ | Wavelength |
| β | Diffraction full width at half-maximum intensity |
| FWHM | Full width at half-maximum intensity |
| $\theta_\beta$ | Angle at maximum intensity and wavelength |
| θ | Angle of incidence of radiation |
| a | Lattice parameter |
| FCC | Face-centered cubic |
| TEM | Transmission electron microscopy |
| EDX | Energy dispersive X-ray measurements |
| ECSA | Electrochemical surface area |
| $I_p$ | Peak current |
| $E_{onset-Ethanol}$ | Ethanol oxidation initiation potential |
| $Q_H$ | Hydrogen desorption charge |
| $m_{Pt}$ | Mass of platinum deposited on the electrode |
| DPP | Decomposition of polymeric precursors |
| AC | Citric acid |
| EG | Ethylene glycol |
| MEA | Membrane electrode assemblies |

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
