# Peer review of "Catalysts of PtSn/C Modified with Ru and Ta for Electrooxidation of Ethanol"

_catalysts, doi:10.3390/catal9030277_

Round 1

Reviewer 1 Report

The paper showed some interesting aspects related to the research on PtSn/C catalyst modified with Ta and Ru for the ethanol electrooxidation. This is a very important issue in the area of fuel cell development. Therefore, I hope that the research of authors in the given field will be continued.

After careful analysis of this document, I suggest that:

- transfer the Material and Methods chapter before the Results chapter.

- add a nomenclature section,

- it would be good to increase the size of the drawings for greater readability.

Author Response

[…] Specific comments:

- transfer the Material and Methods chapter before the Results chapter.

Ans.: We agree with the reviewer´s comment, however the manuscript template of the Catalysts journal recommend the Material Section after the Results Section. All the articles published in the journal present this format.

- add a nomenclature section,

- it would be good to increase the size of the drawings for greater readability.

Ans.: We agree to the viewpoint of the reviewer and we added the nomenclature section in the new manuscript version. Moreover, the size of the drawings were increased for better readability for all the figures present in the new manuscript version.

Reviewer 2 Report

This paper demonstrated new electrocatalyst for the ethanol oxidation using PtSn/C modified with Ru and Ta. The ternary and quaternary catalyst are prepared and investigated using electrochemical methods. However, the difference between nominal and experimental composition is too big. Therefore, it is not easy to identify the tendency between the electrocatalytic ability and the composition. The manuscript is well written and the characterization and experimental is well designed, however, the experimental result is a little complicated to obtain a scientific information of the catalyst.

Comments and questions to reinforce the manuscript:

There is a typo in abstract the “e” between catalyst composition. It seems “and”.

The big difference between nominal and experimental composition should be explained. In addition, which one is more reliable. I don’t understand why the author explained the catalyst based on the nominal (is it just the initial mixing ratio between reagents?) composition not the experimentally observed composition.

From the experimental datum: ECSA, onset potential, or peak current, I couldn’t find any tendency depending on the composition. Especially in some parameters, the ternary catalyst seems better performance than the quaternary catalysts.

In Figure 2 explanation, “the electric double layer loading region (0.15-0.55V)” is mentioned. The selection of wording should be improved because the double layer is exist and keep loading and dis-loading in all potential region.

The (b) and (c) is missing in right figure in Figure 7.

In line 243-244, the value for PtRu/C and PtSnRu/C should be suggested.

It is easy to understand the catalyst if any microscopic image (TEM or SEM) is suggested.

Author Response

- Comments and questions to reinforce the manuscript:

- There is a typo in abstract the “e” between catalyst composition. It seems “and”.

Ans.: We apologize the typographical error. The manuscript was rewritten and typographical errors have been modified, as recommended.

- The big difference between nominal and experimental composition should be explained. In addition, which one is more reliable.

Ans.: The energy dispersive X-ray measurements (EDX) data, Table 1, indicated that the experimental composition of the Ru and Ta modified PtSn/C catalysts were different from the nominal compositions, where the Pt50Sn21Ta25Ru4/C catalyst presented lower platinum loading and Pt75Sn12Ta6Ru7/C catalyst had the highest load of platinum. The difference in the nominal and experimental compositions can be related to the preparation method in the synthesis of the catalysts, thermal decomposition of polymeric precursors, also known by Pechini method. This technique presents a little deviation to the experimental proportions of the different metals when 3 or more components are inserted in the catalyst, fluctuations in relation to the theoretical values are common. [J. Electrochem. Soc. 2013, 160, F853-F858; Int. J. Hydrogen Energy 2011, 36, 3803-38810].

-I don’t understand why the author explained the catalyst based on the nominal […]

-From the experimental datum: ECSA, onset potential, or peak current, I couldn’t find any tendency depending on the composition. Especially in some parameters, the ternary catalyst seems better performance than the quaternary catalysts.

Ans.: Taking into account the reviewer´s relevant comments the figures and in the text the nominal compositions were replaced by the experimental compositions.

Conversely to the comment of the reviewer in the electrochemical measurements, the catalysts showed an increase in the electrocatalytic activity regarding the electrooxidation of ethanol. Where the quaternary Pt75Sn12Ta6Ru7/C catalyst showed better results in cyclic voltammetry in ethanol with lower Eonset-ethanol and also the higher peak current density indicating that the addition of Ta improved the electrocatalytic activity. In the fuel cell test this same composition also presented the best performance compared to the other catalysts synthesized in the paper. When the cell test was conducted with only 2.0 mol L-1 ethanol, it was verified that the lower the Ta content in the catalysts the better its performance. The results indicated that addition of Ta in PtSnRu/C catalysts increased the oxidation capacity of ethanol compared to pure Pt catalysts, in comparison to PtRu/C and PtSnRu/C. However, when the fuel cell test was conducted using ethanol in acid medium the power densities were higher (18-27 mV cm-2) than those found when the test was conducted only in ethanol (12-19 mV cm-2). In addition to the improved performance of the materials in terms of power density it was observed that in acid medium the tendency for the Ta content was reversed.

-In Figure 2 […] The selection of wording should be improved because the double layer is exist and keep loading and dis-loading in all potential region.

Ans.: Taking into account the reviewer´s relevant comment, the text was rewritten.

-The (b) and (c) is missing in right figure in Figure 7.

Ans.: The Figure 7 was removed.

-In line 243-244, the value for PtRu/C and PtSnRu/C should be suggested.

Ans.: Following the suggestion of the reviewer the values found in the literature were added to the text. See new Table 3 in page 12.

-It is easy to understand the catalyst if any microscopic image (TEM or SEM) is suggested.

Ans.: Taking into account the reviewer´s relevant comments, the results and discussion section was rewritten. Moreover, we also added TEM-data to improve the discussion of the manuscript. See Figure 2 in page 5 and Table 1.

Reviewer 3 Report

1.       While the manuscript presents a lot of work on ternary and quaternary catalysts for ethanol oxidation, it lacks a focus on the fundamental mechanism on how Ta and Ru help with ethanol oxidation which is crucial to explain why PtSnTaRu/C is better than PtSnTa/C and PtSn/C. The fundamental aspect should be reflected in motivation (introduction), discussion and conclusion. The characteristics in the material structure and electrochemical property should be relevant to ethanol oxidation, the focus of this work.

2.       Abstract, line 14. Consider rewrite the sentence begins with “Where”.

3.       Introduction, line 47. What do you mean by “not so noble metals”? Please replace it with a professional term. In addition, line 49, Ru, Rh, and Ir are all noble metals and they have equal or higher price than Pt.

4.       Introduction, line 51, line 53, line 55. This part of the introduction is over simplified without addressing the mechanism of doping Pt catalyst with other metals for ethanol oxidation. In addition, a valid hypothesis (motivation) should be included in the introduction regarding why adding Ta or both Ta and Ru to PtSn/C catalyst. How does Ta or Ta and Ru affect the ethanol oxidation step? What aspect can Ta or Ta and Ru improve ethanol oxidation, OH adsorption, C-C bond breaking, de-hydrogenation, etc.?

5.       Line 86-87, how does the expansion of lattice affect ethanol oxidation?

6.       Line 120-123, how does the activation of water relate to ethanol oxidation?

7.       Line 220-222. It is preferred to have a table that summarize these parameters, such onset potentials, mass activity, power density, etc., for comparison with state-of-the-art catalyst, namely PtRu/C, PtSn/C and Pt/C.

8.       Line 256-257. The current variation in chronoamperometry could be from the test equipment and does not equal to the stability for an electrocatalyst. An accelerated-stressed test may be needed to verify the stability.

Author Response

Comments and Suggestions for Authors

1.       While the manuscript presents a lot of work on ternary and quaternary catalysts for ethanol oxidation, it lacks a focus on the fundamental mechanism on how Ta and Ru help with ethanol oxidation which is crucial to explain why PtSnTaRu/C is better than PtSnTa/C and PtSn/C. The fundamental aspect should be reflected in motivation (introduction), discussion and conclusion. The characteristics in the material structure and electrochemical property should be relevant to ethanol oxidation, the focus of this work.

Ans.: Taking into account the reviewer´s relevant comments we added more information about motivation of use the Ta and Ru into the PtSn/C catalysts. See the new introduction in page 4 in the revised manuscript version.

2.       Abstract, line 14. Consider rewrite the sentence begins with “Where”.

Ans.: The sentence of line 14, abstract, was rewritten.

3.       Introduction, line 47. What do you mean by “not so noble metals”? […]

Ans.: The term “not so noble metals” was removed and the sentence was rewritten

4.       Introduction, line 51, line 53, line 55. This part of the introduction is over simplified without addressing the mechanism of doping Pt catalyst with other metals for ethanol oxidation. In addition, a valid hypothesis (motivation) should be included in the introduction regarding why adding Ta or both Ta and Ru to PtSn/C catalyst. How does Ta or Ta and Ru affect the ethanol oxidation step? What aspect can Ta or Ta and Ru improve ethanol oxidation, OH adsorption, C-C bond breaking, de-hydrogenation, etc.?

Ans.:  Studies have shown that the PtSn/C catalyst presents excellent electrocatalytic activity in relation to the ethanol oxidation. The addition of Ru into the PtSn/C catalysts have been also studied for alcohol oxidation. In all these investigations the presence of Ru in the catalyst compositions increase the electrocatalytic activity toward alcohol oxidation. That increase in electrocatalytic activity when Ru is incorporated into the Pt-based catalysts is due to the bi-functional mechanism, where the contaminant (CO), preferentially adsorb to the active sites in Pt is oxidized. When there is another metal, in this case ruthenium and also tin, being more oxidizable than platinum, these metals produce oxygenated species or hydrated oxide acting directly on the oxidation of the contaminant. Another metal that can act as a co-catalyst in platinum-based catalysts is tantalum, papers have been done where binary catalysts of Pt and Ta have improved the tolerance of CO [Electrochim. Acta 2002, 48, 197-204]. Anwar et al. [Int. J. Hydrogen Energy 2017, 42, 30750-30759] used Ta and Ti oxide as support for Pt catalysts which increased the durability compared to pure platinum and maintained their electrocatalytic activity. Masud et al. [J. Power Sources 2012, 220, 399-404] obtained better results for the oxidation of methanol to platinum-based catalysts modified with Ta-oxide. The significant increase in methanol oxidation due to the addition of Ta as co-catalyst together with Pt indicated a crucial role for TaOx in the process. Where the interaction between oxide and metal results in the electron donation of Pt to TaOx thus inducing a positive charge on the surface of Pt [Electrochim. Acta 2010, 55, 2964-2971]. This induced positive charge weakens the bond strength of adsorbed CO at the sites of Pt, thereby increasing its oxidative removal and decreases its surface coverage on the surface of Pt. Another motif in which Ta can improve the performance of Pt catalysts for the oxidation of ethanol is related to the high affinity of their oxides in relation to the OH species, providing species of oxygen at the surface of the catalyst with lower potential compared to pure Pt catalysts. Thus, the presence of Ta maybe favors the bi-functional mechanism and can accelerate the dissociation of H2O molecules to form OHads, facilitating the oxidation of adsorbed alcohols or poisons, thus recovering the active sites of Pt so that the oxidation reaction continues.

5.       Line 86-87, how does the expansion of lattice affect ethanol oxidation?

Ans.: The lattice parameter is an indirect measurement of the distance Pt-Pt bond on metal, thus the solid solution formation (Pt alloys) with Pt and others metals could change the distance of Pt-Pt bond, it which could be also contribute to weaken or modify the C-C bond or CO bond on Pt surface.

6.       Line 120-123, how does the activation of water relate to ethanol oxidation?

Ans.: At low potential (<0.4 V) hydrogen abstraction from ethanol competes with the dissociative adsorption on Pt surface leading to acetaldehyde and CO as the main products. At higher potentials (>0.4 V) the mechanism can be explained from interfacial water activation occurs resulting OH-species which facilitate the formation of high oxidation state compounds (acetic acid, CO2) (J. Appl. Electrochem. 2008, 38, 653).

Performance of PtSnTaRu/C catalysts for the oxidation of ethanol is related to the high affinity of their oxides in relation to the OH species, providing species of oxygen at the surface of the catalyst with lower potential compared to pure Pt catalysts. Thus, the presence of Sn, Ta and/or Ru maybe favors the bi-functional mechanism and can accelerate the dissociation of H2O molecules to form OHads, facilitating the oxidation of adsorbed alcohols or poisons, thus recovering the active sites of Pt so that the oxidation reaction continues.

7.       Line 220-222. It is preferred to have a table that summarize these parameters, such onset potentials, mass activity, power density, etc., for comparison with state-of-the-art catalyst, namely PtRu/C, PtSn/C and Pt/C.

Ans.: Following the suggestion of the reviewer, we added a new Table 3 in page 12, it  which will facilitates the comparison between the catalysts in this paper and others.

8.       Line 256-257. The current variation in chronoamperometry could be from the test equipment and does not equal to the stability for an electrocatalyst. An accelerated-stressed test may be needed to verify the stability.

Ans.: The Figure 7 was removed.

Round 2

Reviewer 2 Report

well revised

Author Response

Does the introduction provide sufficient background and include all relevant references?

ANS.: Conversely to the comment of the reviewer´s, the introduction provide sufficient background and in ours opinion were include all relevant references. See the new introduction in the revised manuscript version. The reviewer could be observe there are twenty seven (27) references in the introduction. However, we include three new references in the revised mnuscript version (refs: 13, 14 and 15): Energies, 2017, 10, 42; Catalysts, 2019, 9, 61 and Catalysts, 2018, 8, 538.

Are the results clearly presented?

ANS.: We have made an effort to improve the results clarity presented in this new revised manuscript version.

Are the conclusions supported by the results?

ANS.: We have made an effort to improve the conclusions based only in the results obtained in this investigations.

Reviewer 3 Report

I thank the authors for considering my comments. While I am satisfied with most of the revision, I do have a few further questions on the revision:

1.       From the revised introduction, it seems like Ru and TaOx share the advantage of affinity to hydroxide and favors the bi-functional mechanism in methanol/ethanol oxidation. With respect to this, since this work incorporated both Ru and Ta in the catalyst, is there any synergistic effect of adding both Ru and Ta?

2.       How does Ru and Ta affect the electronic structure of Pt in this work? The revised introduction states that TaOx induces a positive charge on Pt surface which improves the removal of CO species. Do you observe the same effect as in the literature?

3.       Is Ta present as a metal or oxide in the catalyst of this work? This is not clear, from my view, because XRD results show no sign of Ru or Ta oxide, but Ta oxide is mentioned in the discussions.

4.       The second comment was not addressed. Could the author double check?

Author Response

English language and style are fine/minor spell check required

ANS.: In the revised manuscript version we have made an effort to improve the English.

Are the results clearly presented?

ANS.: We have made an effort to improve the results clarity presented in this new revised manuscript version.

Are the conclusions supported by the results?

ANS.: We have made an effort to improve the conclusions based only in the results obtained in this investigations.

I thank the authors for considering my comments. While I am satisfied with most of the revision, I do have a few further questions on the revision:

1. From the revised introduction, it seems like Ru and TaOx share the advantage of affinity to hydroxide and favors the bi-functional mechanism in methanol/ethanol oxidation. With respect to this, since this work incorporated both Ru and Ta in the catalyst, is there any synergistic effect of adding both Ru and Ta?

ANS.: Although Ta and Ru have similar effects, that is, they share the advantage of affinity to hydroxide and favors the bi-functional mechanism in methanol/ethanol oxidation. They enrich the surface of the catalyst with OHads species and thus facilitate the oxidation of adsorbed species in the Pt sites. Ta also has another advantage, the presence of Ta increase the durability and performance of the catalysts in PEMFC (Int. J. Hydrogen Energy 2017, 42, 30750-30759). Ta was also added in order to obtain catalysts with high tolerance to the intermediates of the ethanol oxidation and CO adsorbed. It was verified the synergic effect related to the addition of Ta and Ru to the PtSn/C catalysts (in our paper), because in the cyclic voltammetry and in the chrono-data the currents obtained are higher than in TaOx modified Pt binary catalysts (J. Power Sources 2012,220, 399-404). Indicating that the presence of Ru and Ta metals increased the ethanol oxidation capacity. However, the ternary PtSnTa/C catalyst gave the best results. Note that the presence the Ta and Ru increased the activity of PtSn/C catalysts.

2. How does Ru and Ta affect the electronic structure of Pt in this work? The revised introduction states that TaOx induces a positive charge on Pt surface which improves the removal of CO species. Do you observe the same effect as in the literature?

ANS.: As can be see, the catalysts exhibit shifted of diffraction peaks to lower values than the 2θ angles of the Pt fcc indicating an increase in the lattice Pt parameter compared to the pure Pt (a = 3.920 nm), probably due to the incorporation of the metals (Sn and/or Ta) in its structure. This expansion of the lattice parameter due to the incorporation of Sn is already well known in the literature (App. Catal. B 2007, 73, 106-115) since its atomic radius (Sn = 151 pm) is greater than that of platinum (Pt = 139 pm). Considering that the atomic radius of Tantalum (Ta = 200 pm) is also greater than that of Pt, this metal can also contribute to the increase of the lattice parameter, leading to the expansion of the crystalline lattice. Moreover, EDX-data showed the presence of oxygen in all compositions: 6.4 at.% for Pt63Sn12Ta25/C, 8.5 at.% for Pt50Sn21Ta25Ru4/C, 7.5 at.% for Pt60Sn16Ta18Ru7/C and 5.8 at.% for Pt75Sn12Ta6Ru6/C. The presence of oxygen in the catalysts suggests the metallic oxide formation (SnOx, RuOx and TaOx). In presence of TaOx occurs the interaction between the d orbitals of platinum with the d orbitals of the oxide resulting in the donation of electrons from Pt to the oxide. This induces a positive charge on the surface of Pt which weakens the binding of CO adsorbed at its sites when in the presence of alcohol. In this way the oxidation of intermediates increases and the surface of the catalyst is regenerated. Based in all the electrochemical results obtained in this investigation, one can infer that the presence of Ta and Ru improve the catalytic activity of PtSn/C-based catalysts due to the improves the removal of CO species.

3. Is Ta present as a metal or oxide in the catalyst of this work? This is not clear, from my view, because XRD results show no sign of Ru or Ta oxide, but Ta oxide is mentioned in the discussions.

ANS.: EDX-data showed the presence of oxygen in all compositions: 6.4 at.% for Pt63Sn12Ta25/C, 8.5 at.% for Pt50Sn21Ta25Ru4/C, 7.5 at.% for Pt60Sn16Ta18Ru7/C and 5.8 at.% for Pt75Sn12Ta6Ru6/C. The presence of oxygen in the catalysts suggests the metallic oxide formation (SnOx, RuOx and TaOx). The EDX-data were include in the new revised manuscript version. (see page 12)

4. The second comment was not addressed. Could the author double check?

ANS.: We apologize the syntax error. All the manuscript was rewritten and syntax errors have been modified, as recommended.